# Interfacial polarization of *in vivo* rat sciatic nerve with crush injury studied via broadband dielectric spectroscopy

**Risa Otagiri[1], Hideki Kawai[2], Masanobu Takatsuka[3], Naoki Shinyashiki[4], Akira Ito[2], Ryosuke Ikeguchi[5], Tomoki Aoyama[2]***

1 Course of Physics, Graduate School of Science, Tokai University, Hiratsuka City, Kanagawa, Japan, 2 Department of Physical Therapy, Human Health Sciences, Graduate school of Medicine, Kyoto University, Kyoto City, Kyoto, Japan, 3 Course of Science and Technology, Graduate School of Science and Technology, Tokai University, Hiratsuka City, Kanagawa, Japan, 4 Department of Physics, School of Science, Tokai University, Hiratsuka City, Kanagawa, Japan, 5 Department of Orthopaedic Surgery, Kyoto University Graduate School of Medicine, Kyoto City, Kyoto, Japan

* aoyama.tomoki.4e@kyoto-u.ac.jp

**Data Availability Statement:** All relevant data are within the manuscript and its Supporting information files.

## Abstract

Electrical stimulation is one of the candidates for elongation-driven regeneration of damaged peripheral nerves. Different organs and tissues have an inherent cell structure and size. This leads to variation in the tissue-specific electrical properties of the frequency of interfacial polarization. Although nervous tissues have a membrane potential, the electrical reaction inside these tissues following electrical stimulation from outside remains unexplored. Furthermore, the pathophysiological reaction of an injured nerve is unclear. Here, we investigated the electrical reaction of injured and non-injured rat sciatic nerves via broadband dielectric spectroscopy. Crush injured and non-injured sciatic nerves of six 12-week-old male Lewis rats were used, 6 days after infliction of the injury. Both sides of the nerves (with and without injury) were exposed, and impedance measurements were performed at room temperature (approximately 25°C) at frequencies ranging from 100 mHz to 5.5 MHz and electric potential ranging from 0.100 to 1.00 V. The measured interfacial polarization potentially originated from the polarization by ion transport around nerve membranes at frequencies between 3.2 kHz and 1.6 MHz. The polarization strength of the injured nerves was smaller than that of non-injured nerves. However, the difference in polarization between injured and non-injured nerves might be caused by inflammation and edema. The suitable frequency range of the interfacial polarization can be expected to be critical for electrical stimulation of injured peripheral nerves.

## Introduction

Injured peripheral nerves can be regenerated by elongation of axons [1]. Poor recovery of damage to peripheral nerves is devastating for several patients, affecting their quality of life. Although patients expect to return to the preinjury conditions, less than half of those

**Funding:** The work of the lead author (T. Aoyama) was supported by the Ministry of Education, Culture, Sports, Science and Technology, Japan (Grant Number: KAKENHI 18H03128).

**Competing interests:** The authors have declared that no competing interests exist.

undergoing nerve repair after peripheral nerve injury achieve good to excellent motor or sensory functionality [1]. Satisfactory motor and sensory outcomes were reported in 52% and 42% of patients, respectively, in a previous study [2]. A 40-year compilation of data showed that after direct nerve coaptation, 20%–40% of patients achieved a good recovery, however, only few patients recovered fully [1, 3]. Furthermore, improvement in patient outcome by surgical reconstruction of the injured nerve continues to pose practical challenges [4], and numerous supportive technologies have been developed to tackle the challenges faced in peripheral nerve repair therapies [1]. Electrical stimulation is one of the candidate therapies for elongating injured nerves [5]. A 20 Hz electrical stimulation was reported to be effective in promoting the expression of the genes for brain-derived neurotrophic factors [6, 7]. Although nervous tissues have a membrane potential [8], the electrical reaction inside the tissue upon electrical stimulation from outside has not yet been investigated. Furthermore, the pathophysiological reaction following electrical stimulation of injured nerves remains unclear.

The measurement of impedance of tissues has been done since a long time [9–11]. For instance, electric impedance of the giant axon of squids was studied in 1930 [12, 13]. Recent developments in technologies for the measurement of impedance allow *in vivo* functional monitoring [14] and imaging of the fascicular organization of peripheral nerves [15]. Broadband dielectric spectroscopy (BDS) offers the broadest frequency range among all the methods for detection of the molecular dynamics in materials. The BDS covers frequencies between 1 μHz and 100 GHz, which correspond to the time window between megaseconds and picoseconds [16]. Furthermore, this frequency range covers the molecular dynamics of solid and liquid materials [17]. For BDS measurements, the material under test is sandwiched between electrodes and an alternate or direct current voltage is applied. The relationship between applied voltage and the induced electric current can be used to obtain the impedance and/or admittance of the material. Next, the complex dielectric constants and/or complex electrical conductivities can be obtained using the impedance or admittance and an electrode with known geometrical capacitance. The rotational motion of molecules with a permanent dipole moment can be observed as dielectric relaxation. Moreover, the translational motion of ionic charges can be observed as conductivity. In materials with special compartments, the translational motion of charges induces polarization within a limited space, which can be observed as interfacial polarization.

Different organs and tissues have inherent cell structures and sizes, which lead to variations in the tissue-specific electrical properties of the frequency of interfacial polarization. In this study, the variation in electrical properties of non-injured and injured peripheral nerves was investigated. To the best of our knowledge, there are few reports investigating the electrical properties of nerves using BDS. In this study, we aimed to clarify the electrical aspect of the pathophysiology of peripheral nerve injuries in rats.

## Materials and methods

### Animals

Six 12-week-old male Lewis rats weighing 250–300 g (Shimizu Laboratory Supplies Co., Ltd, Kyoto, Japan) were purchased and housed in standardized cages with a 12 h light/dark cycle and free access to food and water. All procedures were approved by the Institutional Animal Care and Use Committee of Kyoto University (MedKyo20027). The right sciatic nerve was used as the injury model and the left one was used as the non-injured model to eliminate the effects caused by differences between individuals. A crush injury was created in the right sciatic nerve of anesthetized rats (induced with a mixed anesthetic: 0.15 mg/kg medetomidine, 2 mg/kg midazolam, and 2.5 mg/kg butorphanol) as previously described [18]. This sciatic nerve

crush injury protocol is utilized as a standard peripheral nerve injury model owing to its simplicity and high reproducibility [19, 20]. The sciatic nerve was exposed by a lateral longitudinal incision along the right thigh and was detached from the surrounding tissues. A 2 mm long section of the nerve at the site below the gluteal tuberosity was crushed for 10 s using a needle holder (No. 12501–13, Fine Science Tools Inc., North Vancouver, Canada). The proximal end of the crush injury site was marked using a 9–0 nylon suture (T06A09N20-25, Bear Medic Corporation), and the incision was closed using 4–0 nylon sutures (S15G04N-45, Bear Medic Corporation).

## BDS system

An LCR meter (1 mHz–5.5 MHz, NF corporation, ZM2376) and integrated circuit (IC) clips were used for the measurement of impedance of the sciatic nerves (of rats, assigned names A through F). The measurements were performed at room temperature (approximately 25˚C) using a frequency ($f$) in the range from 100 mHz to 5.5 MHz and an electric potential in the range from 0.100 to 1.00 V and assuming a parallel circuit of resistance ($R_p$) and capacitance ($C_p$). The $f$ dependence of $C_p$ and $R_p$ was obtained through the measurement of impedance. Next, the dielectric constant ($\varepsilon'$), dielectric loss ($\varepsilon''$), resistivity ($\rho$), and electric conductivity ($\sigma$) were obtained using $R_p$ and $C_p$ based on the distances of the clips as well as the diameter of the nerves during each measurement.

Six days after the injury, rats were anesthetized, and both sides of the sciatic nerves were exposed via a posterior longitudinal incision from the gluteal region to the popliteal fossa. The sciatic nerve was then pinched with IC clips at two positions (separated by 3.5–11.6 mm). The right sciatic nerve was used as the injury model and the left one was used as the non-injured model. Impedance of the right sciatic nerve was recorded at three different positions: 1) injury site, at which one IC clip was placed proximal to the injury site, and the other was placed distal to the injury; 2) the proximal site, at which both IC clips were placed proximal to the injury site; and 3) the distal site, at which both IC clips were placed distal to the injury. Spacers were inserted in the clips to prevent nerve damage owing to the pinching pressure of the clips (Fig 1).

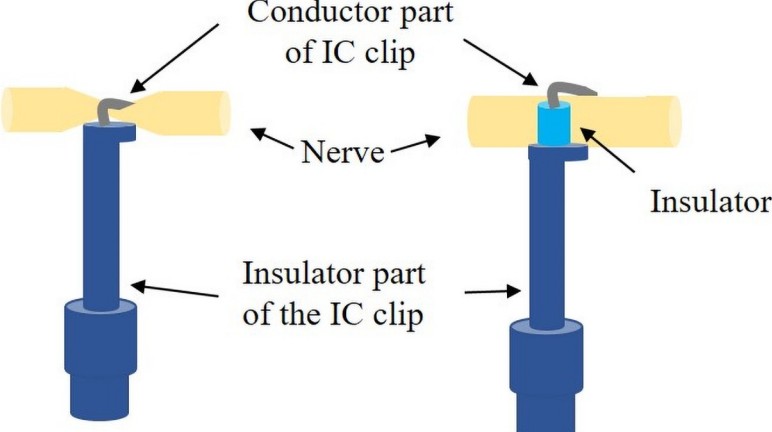

**Fig 1. Schematic illustration of nerves pinched by integrated circuit (IC) clips, with and without spacers.** The left illustration shows the IC clips without spacers, whereas the right illustration shows them with spacers. All impedance measurements were performed with spacers.

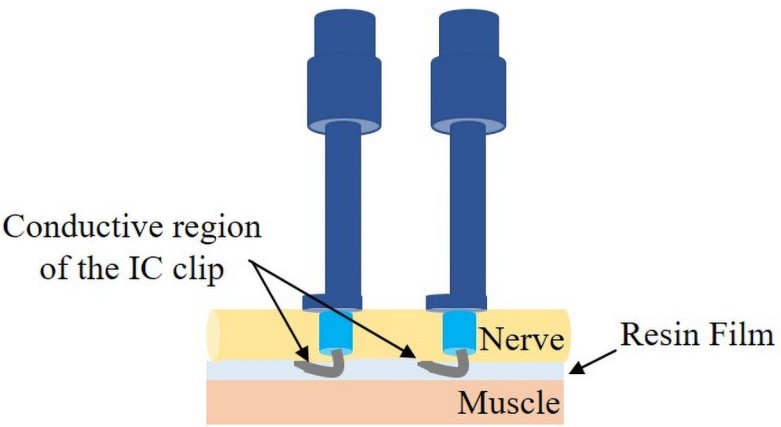

**Fig 2. Schematic illustration of the impedance measurements with resin film between the nerve and integrated circuit (IC) clips.**

In addition, during impedance measurements, a thin resin film was placed between the nerve and muscle to prevent conduction to other areas (e.g., the muscles) through the conductive portions of the IC clips (Fig 2). However, this film was not used for the measurement of impedance of rats A and B. For the measurement of impedance on the injured site of rat A, the selected voltages were 1.0, 0.5, and 0.1 V. There were no differences in the obtained $C_p$ and $R_p$ spectra caused by the differences in the selected voltages. For the other measurement, lower voltages were used (0.5 and 0.1 V) to reduce the change caused by the applied voltages to the nerve for which the impedance was measured. To obtain the geometric capacitances of the nerves, their diameter was measured using a caliper with 0.05 mm graduations. However, because the nerves are deformable, a measurement error of approximately 0.2 mm needs to be considered.

## Histological analysis

Following impedance measurement, a 5 mm long sciatic nerve specimen was dissected from the proximal end of the crush injury site or from the same location of the non-injured nerve. The sections were fixed with 1.44% paraformaldehyde and 1% glutaraldehyde in 0.036 M phosphate buffer (pH = 6.8) at 4°C overnight. Transverse sections were prepared as previously described [18]. Semithin sections were stained with toluidine blue solution, and images were obtained at ×100 magnification using a light microscope (Nikon ECLIPSE 80i, Tokyo, Japan). Ultrathin sections were stained with uranyl acetate and lead citrate, and images were obtained at ×2000 magnification using a transmission electron microscope (Model H-7000, Hitachi High-Technologies, Tokyo, Japan).

## Results and discussion

The different nerve conditions and impedance measurements are presented in S1 Table. The $f$ dependency of $C_p$ and $R_p$ for the measurements presented in S1 Table is shown in Fig 3. When considering frequencies lower than 1.6 MHz, $C_p$ and $R_p$ increased with the decrease in frequency for all data points. While the slope of $C_p$ varies depending on frequency, the $C_p$ slope had a convex shape at approximately 100 Hz (log $f$ = 2) and 30 kHz (log $f$ = 4.5). Thus, the

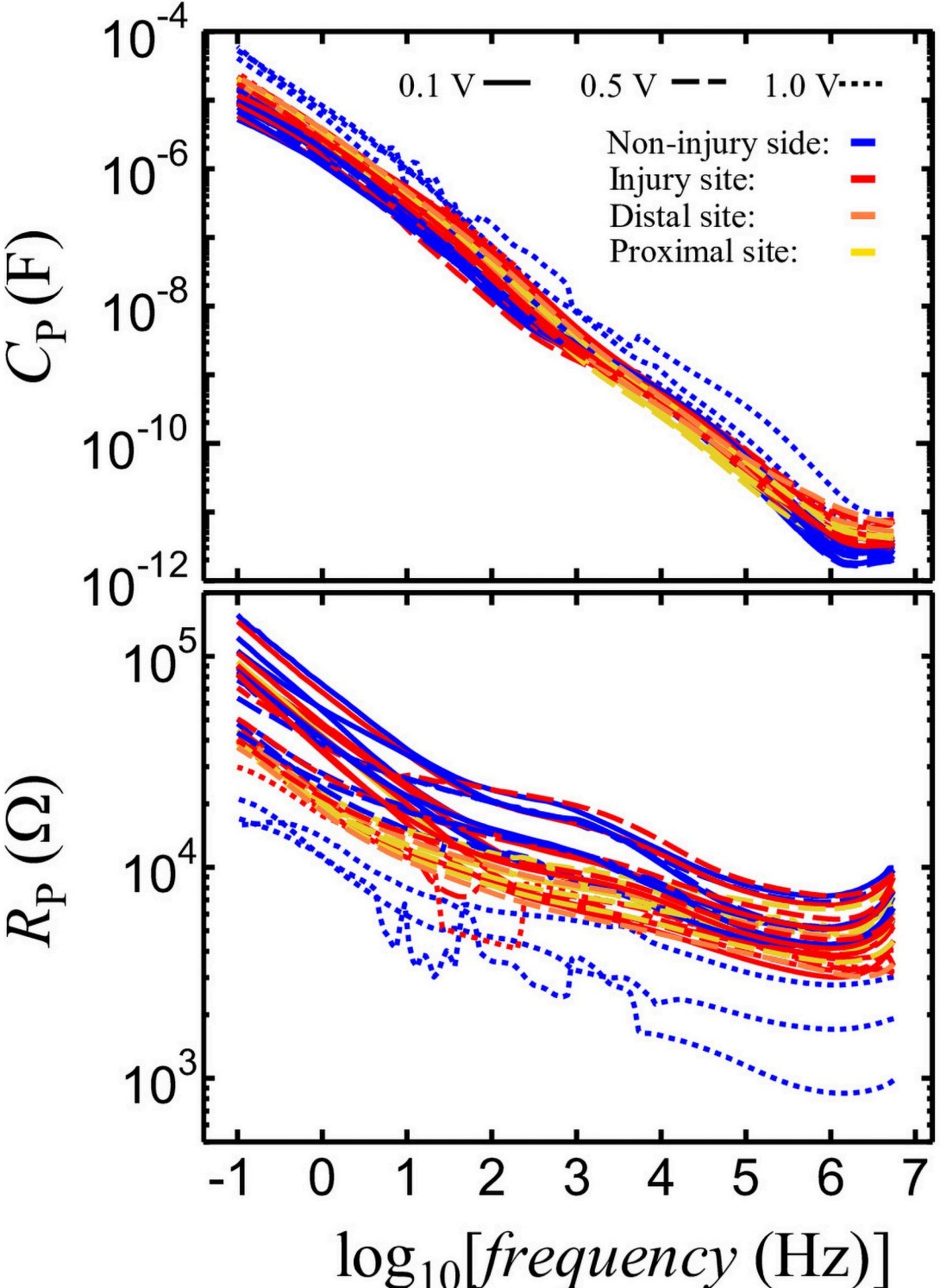

**Fig 3. Frequency dependence of nerve *Cp* and *Rp*.**

steep slope at the higher frequency side of the convex $C_p$ can be attributed to the presence of dielectric relaxation.

Fig 4 shows the dependence of $\varepsilon'$ and $\varepsilon''$ of nerve on frequency. The $\varepsilon'$ and $\varepsilon''$ can be calculated based on the geometric capacitance of the $C_0$ electrodes as follows:

$$\varepsilon' = \frac{C_p}{C_0} \tag{1}$$

$$\varepsilon'' = \frac{1}{2\pi f C_0 R_p} \tag{2}$$

Here, $C_0$ was calculated using Eq (3).

$$C_0 = \varepsilon_0 \frac{S}{d} \tag{3}$$

where $S$ represents the sectional area of the nerve and $d$ is the distance between the IC clips. $S$ can be calculated as follows:

$$S = \pi \left(\frac{\Phi}{2}\right)^2 \tag{4}$$

where $\Phi$ is the nerve diameter. The changes in slope were also observed in the frequency dependence of $\varepsilon'$. For all samples, the dielectric relaxation process, which can be recognized as a steep slope area, was observed in the frequency range from 10 kHz to 1 MHz. At frequencies higher than 1.6 MHz, the increase in $\varepsilon'$ observed following an increase in frequency can be attributed to an error in our measurements because of the effects of the distribution constant circuit at high frequencies.

The frequency dependence of nerve resistivity ($\rho$) and conductivity ($\sigma$) is shown in Fig 5, where $\rho$ is calculated as:

$$\rho = R_p \frac{S}{d} \tag{5}$$

Here, $\sigma$ is the reciprocal of $\rho$, as:

$$\sigma = \frac{1}{\rho} \tag{6}$$

For all the samples, the steep aspect of the $\sigma$ and $\rho$ slopes is induced by the occurrence of dielectric relaxation at frequencies higher than 1 kHz. Within the 1 kHz to 100 Hz range, the slopes are gentle, whereas they are steep for frequencies below 100 Hz. The values for frequencies between 1 kHz and 100 Hz correspond to direct current (DC) and electrical conduction by mobile ionic substances. However, electrode polarization occurred below 100 Hz, whereas the ionic substances that contributed to DC conduction were immobilized at the interfaces of the sample and electrodes (IC clips).

IC clips were used as electrodes for the measurement of impedance. The length of nerve between the two IC clips was obtained with an error of approximately 10% or less. However, the areas of the electrodes were not accurately obtained because of the use of IC clips. Generally, the structures of electrodes used for dielectric measurements are composed of parallel plates or cylindrical cells, the dimensions of which can be determined accurately. Thus, the

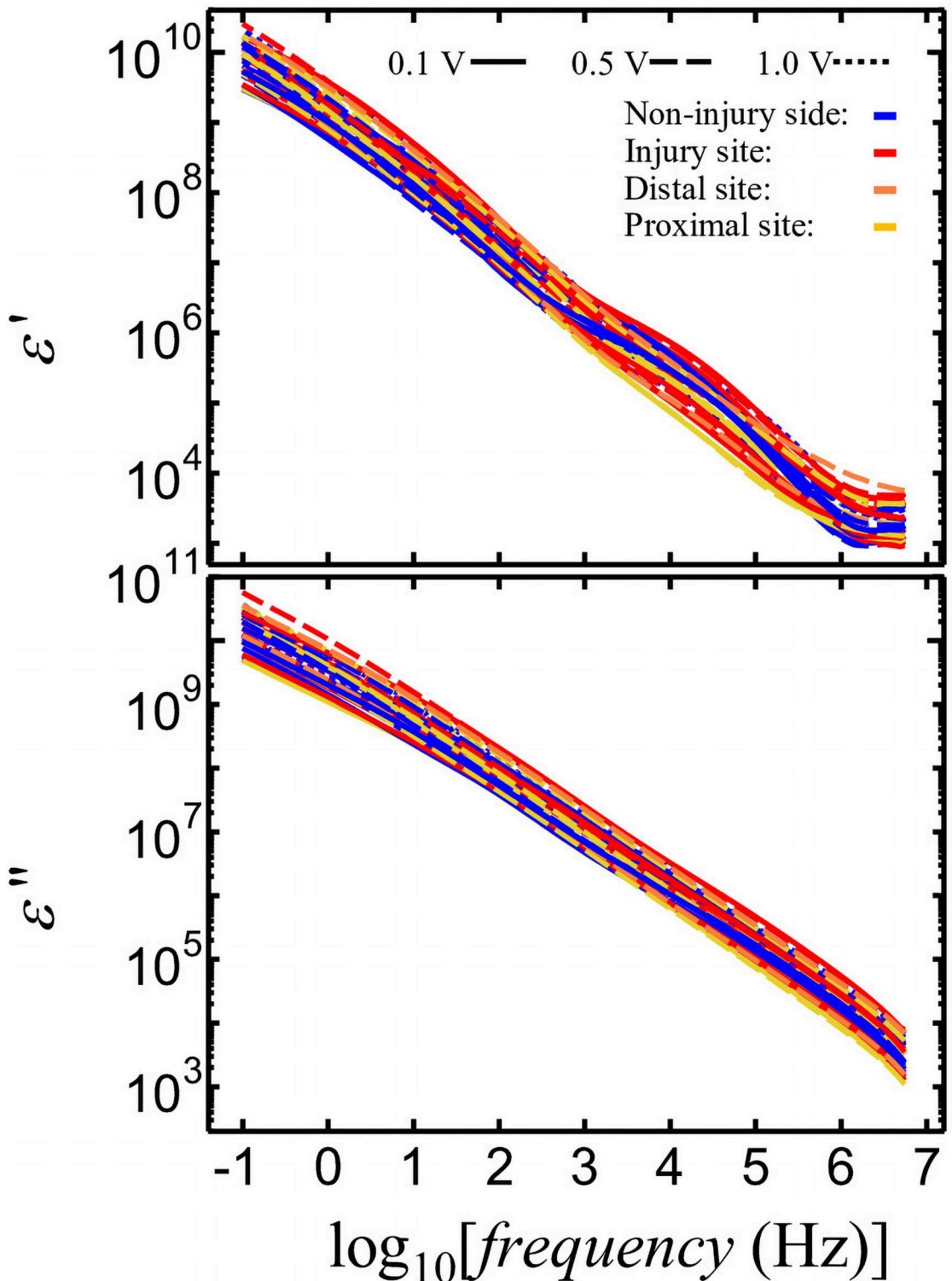

**Fig 4. Frequency dependence of nerve dielectric constant, $\varepsilon'$, and dielectric loss, $\varepsilon''$.**

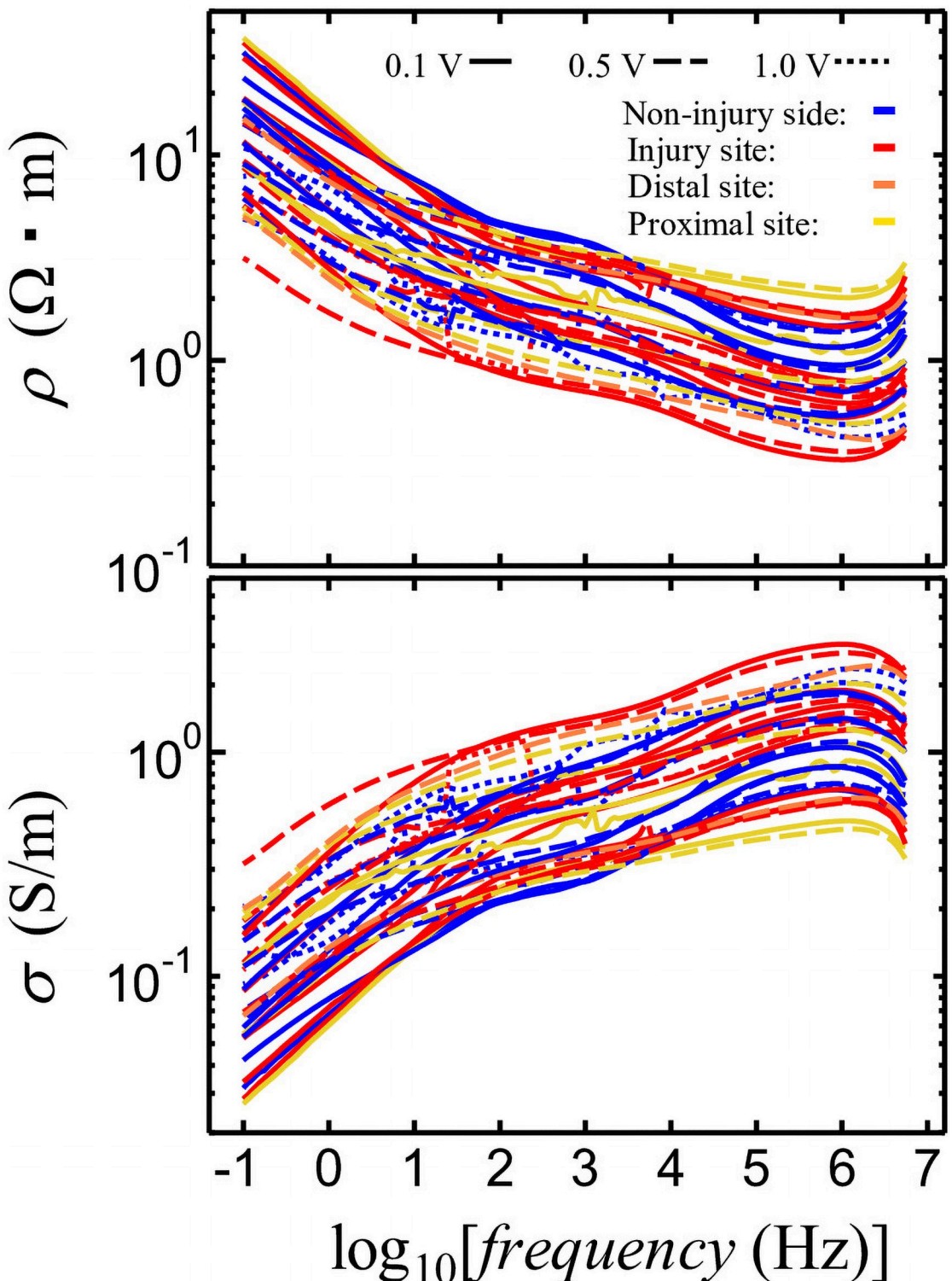

**Fig 5. Frequency dependence of nerve resistivity ($\rho$) and electric conductivity ($\sigma$).**

absolute values of $\varepsilon'$, $\varepsilon''$, $\rho$, and $\sigma$ are unreliable. To address this issue, $\varepsilon'$ was normalized by $\varepsilon'$ ($f$ = 1.6 MHz), by dividing the $\varepsilon'$ ($f$) at all measured frequencies by their specific $\varepsilon'$ ($f$ = 1.6 MHz), to identify the change in $\varepsilon'$ value from 10 kHz to 1.6 MHz, which corresponds to the appearance of interfacial polarization in nerves. Here, following normalization, $\varepsilon'$ is represented as $\varepsilon'_{norm}$ ($f$). We selected an $\varepsilon'$ frequency of 1.6 MHz for normalization because there was an error for $\varepsilon'$ frequencies higher than 1.6 MHz. Furthermore, the dielectric constant of the nerve at frequencies higher than 1 MHz might be attributable to the dielectric constant of water, which is expected to be identical for all measurements. Thus, the variation in nerve $\varepsilon'$ at 1.6 MHz (Fig 4) might be produced by the error in the measurement of the geometrical capacitance of the electrode comprising the IC clips. Therefore, the $\varepsilon'_{norm}$ ($f$) using $\varepsilon'$ ($f$ = 1.6 MHz) might be useful for comparing the strengths of interfacial polarization between 10 kHz and 1.6 MHz. The frequency dependence of $\varepsilon'_{norm}$ ($f$) is shown in Fig 6. The non-injured nerves (**blue lines**) show the largest normalized $\varepsilon'$ values at 10 kHz followed by the normalized $\varepsilon'$ (10 kHz) values measured at the injury site (**red lines**), the proximal site (**yellow lines**), and the distal site (**orange lines**).

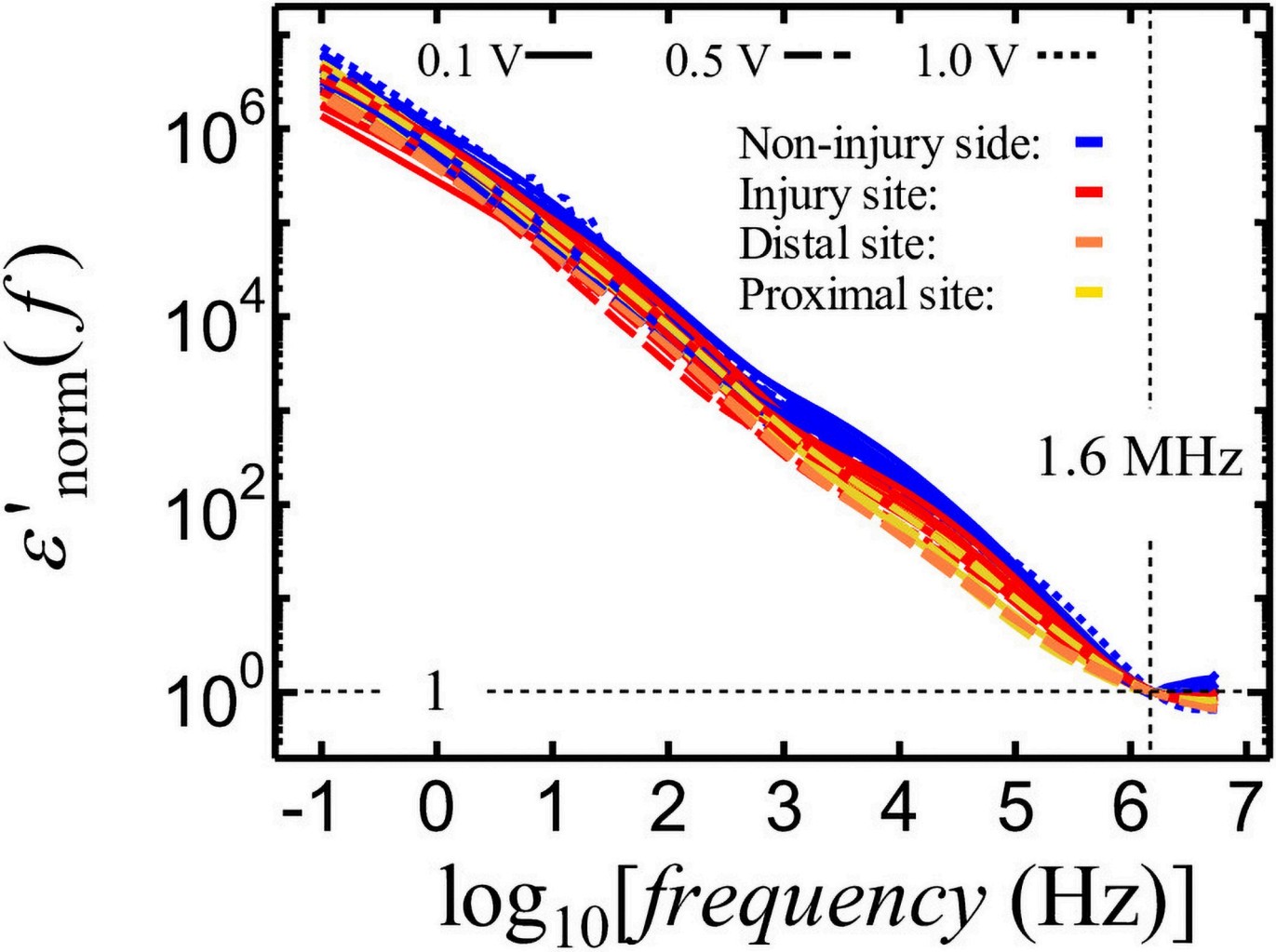

**Fig 6. Frequency dependence of nerve $\varepsilon'$ normalized.**

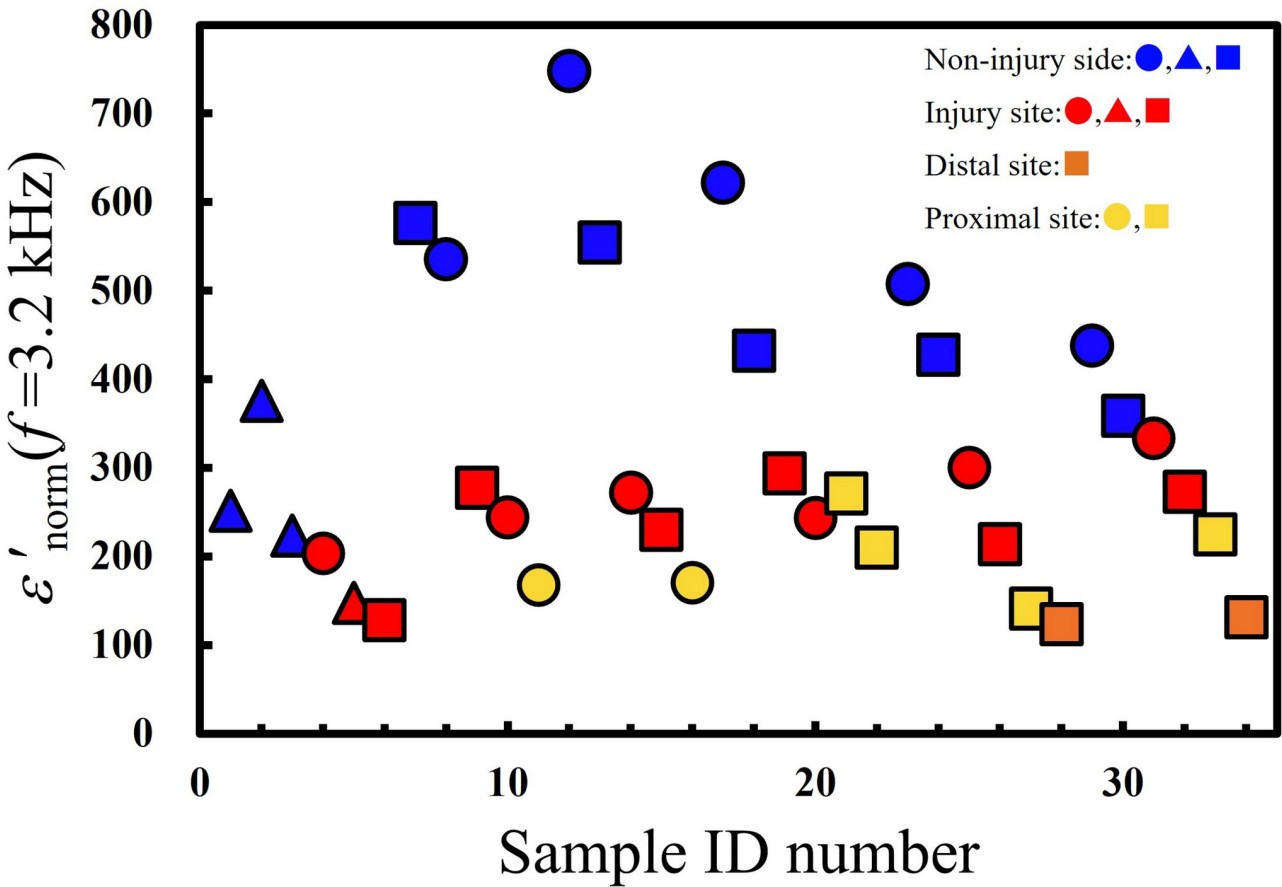

**Fig 7. Plots of $\varepsilon'$ at 3.2 kHz normalized by $\varepsilon'$ ($f$ = 1.6 MHz) of the nerves.** The shapes indicate the applied voltage as 0.1 V (circles), 0.5 V (squares), and 1.0 V (triangles).

To examine the characteristics of interfacial polarization within the 3.2 kHz–1.6 MHz frequency range, $\varepsilon'_{norm}$ ($f$ = 3.2 kHz) was obtained as follows:

$$\varepsilon'_{norm}(f = 3.2\ kHz) = \frac{\varepsilon'(f = 3.2\ kHz)}{\varepsilon'(f = 1.6\ MHz)}. \tag{7}$$

The values of $\varepsilon'_{norm}$ ($f$ = 3.2 kHz) for all measured nerves are shown in Fig 7. The descending order of $\varepsilon'_{norm}$ ($f$ = 3.2 kHz) values comprised the non-injured nerves (**blue lines**), the injured sites (**red lines**), and the proximal and distal sites (**yellow** and **orange lines**, respectively).

An injury reduces the strength of the interfacial polarization within this frequency range. Interfacial polarization occurs in the material with a confined structure. The interfaces in the nerves confine the space for translational motion of the ionic material in nerves. The large amount of mobile ionic materials and/or the presence of a clearer interface contribute to enhancing the strength of interfacial polarization. We inferred that the reduction in strength might be induced by the reduction in interfaces, which are represented by the nerve membrane. Based on the presence of interfacial polarization, the applied alternate voltage effectively induced the translational motion of the ions in the nerve. We hypothesize that electrical

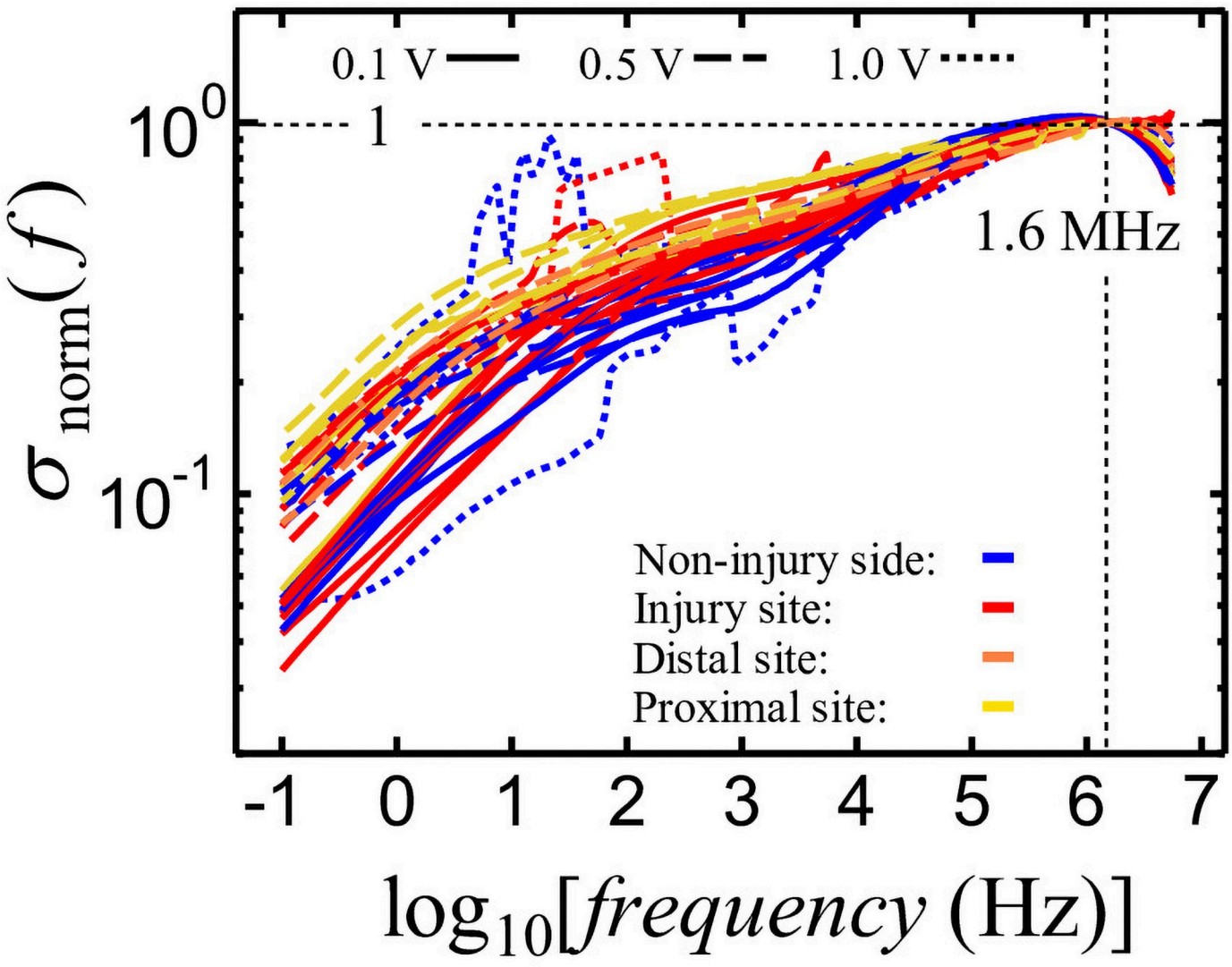

**Fig 8. Frequency dependence of nerve $\sigma$ normalized by $\sigma$ ($f$ = 1.6 MHz).**

stimulation using an alternating current with frequency in the range of interfacial polarization may affect nerve regeneration.

Here, $\sigma$ was normalized using $\sigma$ ($f$ = 1.6 MHz) to compare the variation in $\sigma$ between samples stimulated within the 1.6 MHz to 1 kHz range. The frequency dependence of the normalized $\sigma$ is shown in Fig 8.

To characterize the contribution of the interfacial polarization of the nerve, the $\sigma$ at 1 kHz, which is the frequency corresponding to approximately the middle of the gentle slope in the 100 Hz–10 kHz range, was normalized by $\sigma$ at 1.6 MHz as:

$$\sigma_{norm}(f = 1\ kHz) = \frac{\sigma(f = 1\ kHz)}{\sigma(f = 1.6\ MHz)} \tag{8}$$

The obtained $\sigma_{norm}$ ($f$ = 1 kHz) is shown in Fig 9. In case of $\sigma_{norm}$ ($f$ = 1 kHz), the larger contribution of interfacial polarization represents a smaller value of $\sigma_{norm}$ ($f$ = 1 kHz). The

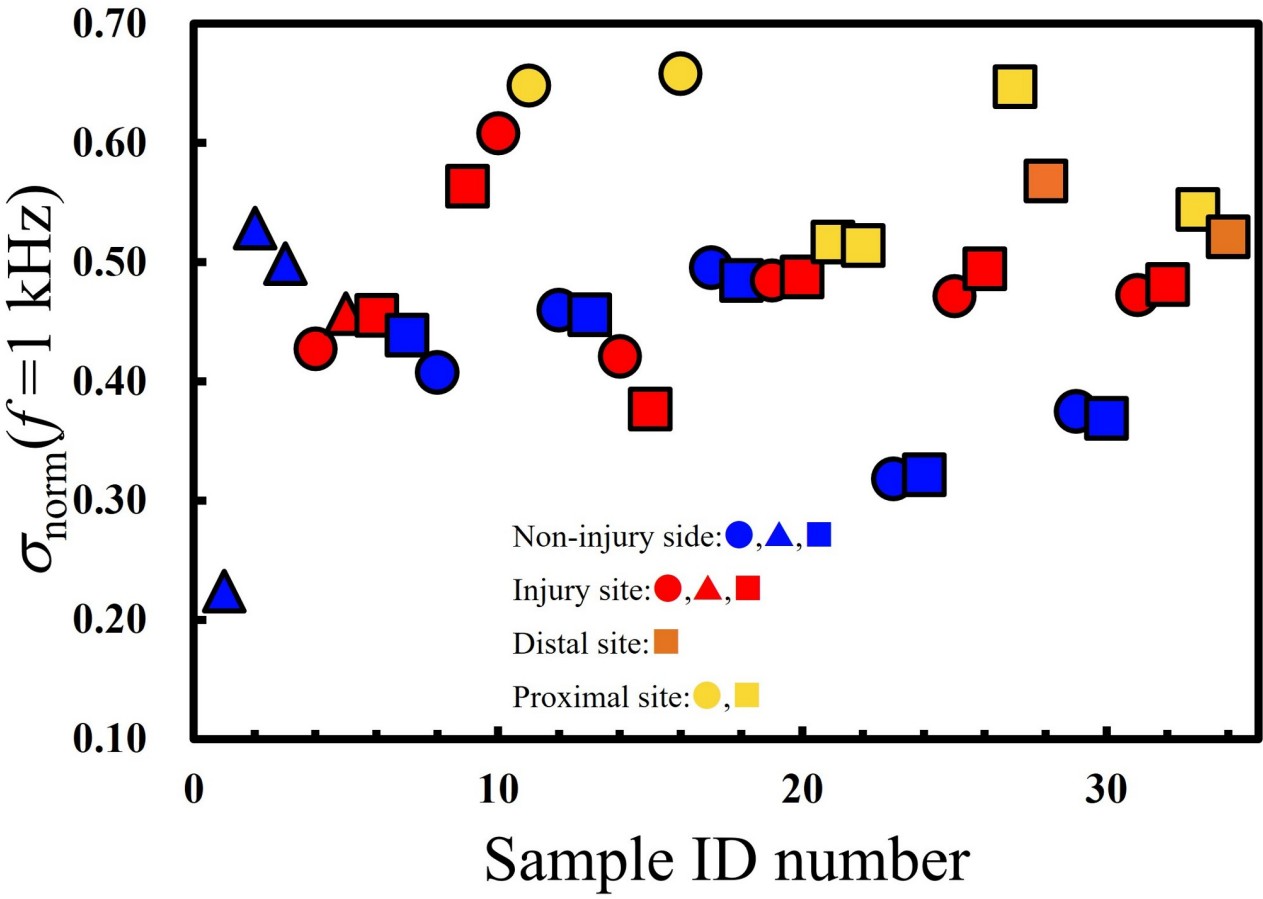

**Fig 9. Plots of σ at 1 kHz normalized by σ (*f* = 1.6 MHz) of the nerves.** The shapes of the plots indicate the applied voltage as 0.1 V (circles), 0.5 V (squares), and 1.0 V (triangles).

smallest values of $\sigma_{\text{norm}}$ ($f$ = 1 kHz) were observed for non-injured nerves (**blue**), followed by those for injured nerves measured at the injury sites (**red**), and the proximal and distal sites (**yellow** and **orange**, respectively). The differences between the injured and non-injured nerves are rather unclear if compared with those observed in the $\varepsilon'_{\text{norm}}$ ($f$ = 3.2 kHz).

Measurement of impedance within the 100 mHz–5.5 MHz frequency range was performed on live injured and non-injured rat nerves to study the frequency dependence of $C_p$, $R_p$, $\varepsilon'$, $\varepsilon''$, $\rho$, and $\sigma$. The dielectric relaxation process of interfacial polarization is expected to originate from ion transportation-induced polarization within nerves and can be observed in the 3.2 kHz to 1.6 MHz frequency range. Although the absolute $\varepsilon'$ value could not be obtained because of the use of IC clips, the $\varepsilon'$ normalized by the $\varepsilon'$ at 1.6 MHz value could be used. The largest $\varepsilon'_{\text{norm}}$ ($f$ = 3.2 kHz) values were observed for the non-injured nerves, followed by those for the injured nerves measured at the injury site, the proximal, and distal sites. We show that nerve injuries reduce the strength of the interfacial polarization.

The active potential of nerves has a 1 ms pulse width and it corresponds to 1 kHz. The frequency range of the interfacial polarization was higher than that of the active potential. Therefore, if the observed interfacial polarization originates from the polarization of ions around the nerve membrane, a higher stimulus rate would affect nerve activation. The stimulation with an

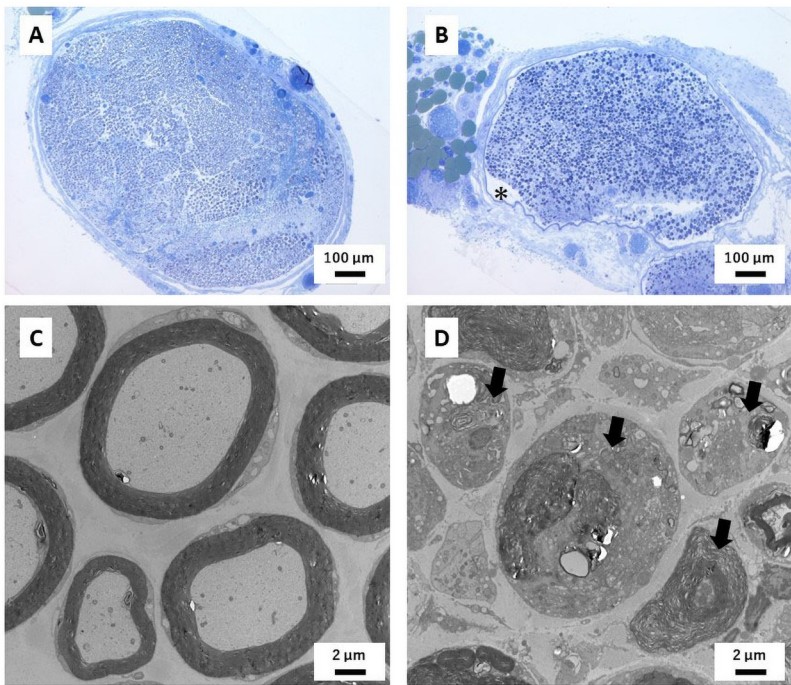

**Fig 10. Representative images of transverse sections of the sciatic nerve.** (A, B) Semithin sections of non-injured (A) and injured (B) sciatic nerve. Following injury, spaces between the perineurium and the nerve fascicle were enlarged (asterisk). (C, D) Ultrathin sections of non-injured (C) and injured (D) sciatic nerves. The axons distal to injured site are degenerated (Arrow).

alternate voltage and frequency similar to the interfacial polarization is expected to effectively induce polarization in nerves.

Following peripheral nerve injury, inflammatory cells, such as macrophages, are infiltrated and axons distal to the injury site degenerate and are removed, whereas axons of the non-injured nerve or those proximal to the injury remain normal [21]. Additionally, crush injuries induce edema [22]. After injury, spaces resembling edema between the perineurium and the nerve fascicle became enlarged (Fig 10B asterisk) and axons degenerate (Fig 10D). Inflammation induces the secretion of cytokines, such as TNF and IL-1 [23]. Such inflammatory cytokines recruit inflammatory cells into the location of the injured nerve and induce axonal degeneration [24]. Such pathological changes affect the composition of the nerve. Although the axons proximal to the injured site should not degenerate, the values of $\varepsilon'_{norm}$ (f = 3.2 kHz) at the proximal site are similar to those at the injured site or distal site (Fig 7). Inflammation and edema affect impedance [25], which might explain the similarities between the proximal site and the injured site [26, 27]. These results indicate that changes in impedance caused by injury are because of inflammation or edema rather than axonal degeneration.

Impedance analysis is practically applicable in many fields, especially in clinical and health-care fields. The assessment of body composition is useful for the diagnosis of sarcopenia [28, 29], and the mucosal impedance is useful for evaluating esophageal disease [30]. Recently, applied impedance technologies have been used to evaluate cells [31] and detect bacteria [32]. The organs, tissues, and cells have their own protein compositions and characteristic electric properties. The developed *in vivo* impedance measurement system for understanding the

pathophysiology of peripheral nerve injuries using BDS may provide novel insights for developing new therapeutics for nerve regeneration.

## Conclusion

In this study, we measured the *in vivo* impedance of injured and non-injured nerves using BDS. The dielectric relaxation process of interfacial polarization, which is expected to originate because of the polarization by ions around the nerve membrane, was observed in the 3.2 kHz to 1.6 MHz frequency range. This implies that a stimulus rate higher than 10 kHz would affect nerve activation. The strength of injured nerve interfacial polarization was smaller than that observed in nerves without an injury. This reduction in strength might be caused by the reduction in interfaces, which are represented by the nerve membrane. These results also illustrate the capacity of BDS for use as a tool to understand the electrical aspect of the pathophysiology of nerve injuries.

## Supporting information

**S1 Table. Samples informations.**
(DOCX)

## Acknowledgments

We are grateful to staff members at Tokai University and Kyoto University for their support.

## Author Contributions

**Conceptualization:** Naoki Shinyashiki, Tomoki Aoyama.

**Data curation:** Risa Otagiri, Hideki Kawai, Masanobu Takatsuka, Tomoki Aoyama.

**Formal analysis:** Masanobu Takatsuka, Naoki Shinyashiki.

**Funding acquisition:** Naoki Shinyashiki, Ryosuke Ikeguchi, Tomoki Aoyama.

**Investigation:** Naoki Shinyashiki, Akira Ito.

**Methodology:** Risa Otagiri, Hideki Kawai, Masanobu Takatsuka, Naoki Shinyashiki.

**Project administration:** Risa Otagiri, Hideki Kawai, Masanobu Takatsuka, Naoki Shinyashiki, Akira Ito.

**Resources:** Risa Otagiri, Hideki Kawai.

**Software:** Risa Otagiri.

**Supervision:** Naoki Shinyashiki, Ryosuke Ikeguchi, Tomoki Aoyama.

**Validation:** Ryosuke Ikeguchi.

**Writing – original draft:** Risa Otagiri.

**Writing – review & editing:** Hideki Kawai, Naoki Shinyashiki, Akira Ito, Ryosuke Ikeguchi, Tomoki Aoyama.

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
