## [Decision Letter · Decision Letter 0]

14 Apr 2021

PONE-D-21-05156

Interfacial polarization of in vivo rat sciatic nerve with crush injury studied via broadband dielectric spectroscopy

PLOS ONE

Dear Dr. Aoyama,

Thank you for submitting your manuscript to PLOS ONE. After careful consideration, we feel that it has merit but does not fully meet PLOS ONE’s publication criteria as it currently stands. Therefore, we invite you to submit a revised version of the manuscript that addresses the points raised during the review process.

We look forward to receiving your revised manuscript.

Kind regards,

Rosanna Di Paola, MD

Academic Editor

PLOS ONE

Journal Requirements:

PLOS requires an ORCID iD for the corresponding author in Editorial Manager on papers submitted after December 6th, 2016. Please ensure that you have an ORCID iD and that it is validated in Editorial Manager. To do this, go to ‘Update my Information’ (in the upper left-hand corner of the main menu), and click on the Fetch/Validate link next to the ORCID field. This will take you to the ORCID site and allow you to create a new iD or authenticate a pre-existing iD in Editorial Manager. Please see the following video for instructions on linking an ORCID iD to your Editorial Manager account: https://www.youtube.com/watch?v=_xcclfuvtxQ

In your Data Availability statement, you have not specified where the minimal data set underlying the results described in your manuscript can be found. PLOS defines a study's minimal data set as the underlying data used to reach the conclusions drawn in the manuscript and any additional data required to replicate the reported study findings in their entirety. All PLOS journals require that the minimal data set be made fully available. For more information about our data policy, please see http://journals.plos.org/plosone/s/data-availability.

Reviewers' comments:

Reviewer's Responses to Questions

**Comments to the Author**

1. Is the manuscript technically sound, and do the data support the conclusions?

Reviewer #1: Yes

Reviewer #2: Partly

2. Has the statistical analysis been performed appropriately and rigorously? 

Reviewer #1: Yes

Reviewer #2: Yes

3. Have the authors made all data underlying the findings in their manuscript fully available?

Reviewer #1: Yes

Reviewer #2: Yes

4. Is the manuscript presented in an intelligible fashion and written in standard English?

Reviewer #1: Yes

Reviewer #2: Yes

5. Review Comments to the Author

Reviewer #1: The submission from Rita Otagiri et al. reports that the strength of injured nerve interfacial polarization was smaller than that observed in nerves without an injury. Moreover, authors suggest that BDS could be used as a tool to understand the electrical aspect of the pathophysiology of nerve injury. The study is interesting and well structured. However, there are some corrections.

Minor comments:

1. The authors should add some more current references about sciatic nerve with crush injury (doi: 10.3390/ijms21103509; doi: 10.1186/s12974-018-1303-5)

2. The authors should include in the materials and methods section the experimental groups they analyzed during the experiment.

3. The authors should better check the manuscript for any typographical errors.

Reviewer #2: This study investigated the mechanism underlining the interfacial polarization of in vivo rat sciatic nerve with crush injury. The rational behind the experiment was clear and straight forward. The manuscript is almost well written.

There are some minor grammar issues that should be fixed in order to aid the accessibility of the results to

the reader.

6. PLOS authors have the option to publish the peer review history of their article (what does this mean?). If published, this will include your full peer review and any attached files.

Reviewer #1: No

Reviewer #2: No

---

## [Author Response · Author response to Decision Letter 0]

29 Apr 2021

Response to Reviewers

We sincerely thank the reviewers for their thoughtful comments, which helped us improve our manuscript. The manuscript has been revised to address the reviewers’ concerns. Revised Pages mentioned below are from the revised manuscript with red words.

[Comment 1 ]

The authors should add some more current references about sciatic nerve with crush injury (doi: 10.3390/ijms21103509; doi: 10.1186/s12974-018-1303-5)

[Response 1 ]

Thank you for the suggestion. We have cited the two references (#19 and 20) regarding sciatic nerve with crush injury, as suggested by you.

[Comment 2 ]

The authors should include in the materials and methods section the experimental groups they analyzed during the experiment.

[Response 2 ]

Thank you for the suggestion. As suggested, we have added a sentence describing the experimental groups analyzed in the study under the subsection titled “Animals.” 

[Comment 3]

The authors should better check the manuscript for any typographical errors.

[Response 3]

We apologize for the typographical errors. The manuscript has been carefully proofread for language and spellings by a professional English language editing company.

---

## [Editor Report · Decision Letter 1]

19 May 2021

Interfacial polarization of in vivo rat sciatic nerve with crush injury studied via broadband dielectric spectroscopy

PONE-D-21-05156R1

Dear Dr. 

We’re pleased to inform you that your manuscript has been judged scientifically suitable for publication and will be formally accepted for publication once it meets all outstanding technical requirements.

Kind regards,

Rosanna Di Paola, MD

Academic Editor

PLOS ONE
---

## [Editor Report · Acceptance letter]

21 May 2021

PONE-D-21-05156R1 

Interfacial polarization of *in vivo* rat sciatic nerve with crush injury studied via broadband dielectric spectroscopy 

Dear Dr. Aoyama:

I'm pleased to inform you that your manuscript has been deemed suitable for publication in PLOS ONE. Congratulations! Your manuscript is now with our production department. 

Kind regards, 

on behalf of

Dr. Rosanna Di Paola 

Academic Editor

PLOS ONE